# Prehospital acute life-threatening cardiovascular disease in elderly: an observational, prospective, multicentre, ambulance-based cohort study

Carlos del Pozo Vegas,[1,2] Daniel Zalama-Sánchez,[1] Ancor Sanz-Garcia [iD],[3] Raúl López-Izquierdo,[2,4,5] Silvia Sáez-Belloso,[2,6] Cristina Mazas Perez Oleaga,[7,8,9] Irma Domínguez Azpíroz,[7,10,11] Iñaki Elío Pascual,[7,11,12] Francisco Martín-Rodríguez[2,6]

For numbered affiliations see end of article.

**Correspondence to**
Dr Ancor Sanz-Garcia;
ancor.sanz@gmail.com

## ABSTRACT

**Objective** The aim was to explore the association of demographic and prehospital parameters with short-term and long-term mortality in acute life-threatening cardiovascular disease by using a hazard model, focusing on elderly individuals, by comparing patients under 75 years versus patients over 75 years of age.

**Design** Prospective, multicentre, observational study.

**Setting** Emergency medical services (EMS) delivery study gathering data from two back-to-back studies between 1 October 2019 and 30 November 2021. Six advanced life support (ALS), 43 basic life support and five hospitals in Spain were considered.

**Participants** Adult patients suffering from acute life-threatening cardiovascular disease attended by the EMS.

**Primary and secondary outcome measures** The primary outcome was in-hospital mortality from any cause within the first to the 365 days following EMS attendance. The main measures included prehospital demographics, biochemical variables, prehospital ALS techniques used and syndromic suspected conditions.

**Results** A total of 1744 patients fulfilled the inclusion criteria. The 365-day cumulative mortality in the elderly amounted to 26.1% (229 cases) versus 11.6% (11.6%) in patients under 75 years old. Elderly patients (≥75 years) presented a twofold risk of mortality compared with patients ≤74 years. Life-threatening interventions (mechanical ventilation, cardioversion and defibrillation) were also related to a twofold increased risk of mortality. Importantly, patients suffering from acute heart failure presented a more than twofold increased risk of mortality.

**Conclusions** This study revealed the prehospital variables associated with the long-term mortality of patients suffering from acute cardiovascular disease. Our results provide important insights for the development of specific codes or scores for cardiovascular diseases to facilitate the risk of mortality characterisation.

## INTRODUCTION

Cardiovascular diseases represent the leading cause of prehospital care, involving a surprising number of unplanned hospitalisations and

## STRENGTHS AND LIMITATIONS OF THIS STUDY

⇒ Here, we present a prospective, multicentre, observational study.

⇒ We explored all adult patients suffering from acute life-threatening cardiovascular disease attended by the emergency medical services.

⇒ We present a relevant sample size with a reduced loss-to-follow-up rate.

⇒ As an observational study, this could entail a selection bias.

⇒ Some doubts could arise with prehospital symptoms due to the lack of complementary tests.

sudden unexplained mortality.[1] Emergency medical services (EMS) must handle this overwhelming patient workload quickly and efficiently, following clinical guidelines and under recognised training, for example, basic and advanced cardiac life support (BCLS and ACLS).[2 3]

The setup and implementation of specific detection codes for life-threatening conditions, for example, cardiorespiratory arrest, ST-elevation coronary syndrome and stroke, are a well-established procedure in health systems, and EMS plays an active role in detection, emergency critical care and assisted transfer to a suitable hospital.[4] Predefined standard protocols should be applied in diseases with clear symptomatology or well-defined guiding symptoms, and in general, no major operational problems are encountered. The handicap for EMS providers consists of the identification on-scene or *en* route of patients with acute life-threatening cardiovascular disease that is apparently masked. Early warning scores and point-of-care testing can provide an effective support tool to help at critical junctures in

the complicated decision-making process.[5 6] However, the identification of high-risk subjects is challenging, particularly when distracting factors such as comorbidities or age come into play.[7]

EMS has increasingly switched types of patients, with older adults becoming a major focus area of care.[8] Defining the concept of older adults is complex, as there are different categories and timelines, and there is no standard criterion to say that an older person is considered elderly.[9] However, there is a general acceptance that persons over 65 years of age are considered elderly, and persons over 75 years of age are classified as late elderly. On the other hand, life expectancy and ageing control have improved considerably, with age-related comorbidities usually appearing later, which is the main reason why the 75-year age limit has been selected to differentiate the cohorts of the present work.[10] Among other reasons, falls, drug-taking mistakes and exacerbations of chronic pathologies are more frequent in elderly individuals, that is, EMS providers can attend to cases of older adults with comorbidities and multimedication. In particular, cardiovascular diseases constitute the first cause of emergency appointments and inpatient hospitalisation in older adults, affecting both men and women.[11] Life expectancy has increased significantly, meaning that older adults with atrial fibrillation, acute coronary syndrome, congestive heart failure, valvular heart disease and other cardiovascular processes are becoming increasingly prevalent.[12] Complications related to ageing can hamper anamnesis and clinical examination and sometimes disrupt responsiveness mechanisms, for example, 20% of older adults exhibit atypical symptoms of acute coronary syndrome.[13] Additionally, coexisting comorbidities may trigger interactions in the cardiovascular system, such as anaemia, chronic kidney disease or diabetes.[14] Comorbidities naturally rise sharply with age and necessitate an appropriate analysis. Nonetheless, at 75 years old, cardiovascular pathology, in particular, showed considerable progression with significant increases in comorbidities, the emergence of additional diseases and the exacerbation of already prevalent pathologies. The group of elderly individuals over 75-year-old constitutes a cluster of special follow-ups that could be especially worthwhile for an in-depth characterisation.[10 15]

The purpose of the study was to explore the association of demographic and prehospital parameters with short-term and long-term mortality in acute life-threatening cardiovascular disease by using a hazard model. In particular, we focused on the elderly by comparing two age cohorts: under 75 years vs over 75 years.

## METHODS
### Study design
This prospective, multicentre, observational, EMS delivery analysis gathers inputs obtained from two back-to-back studies, 'prehospital identification of prognostic biomarkers in time-dependent diseases' (HITScore) and 'identification of biomarkers of clinical-risk deterioration in prehospital care' (preBIOs).

This study was reported according to STrengthening the Reporting of OBservational studies in Epidemiology (online supplemental data P3)[16] and complies with the Declaration of Helsinki.

### Study setting
The study was conducted in four Spanish provinces (Burgos, Salamanca, Segovia and Valladolid), enrolling uninterrupted adults (>18 years) with syndromic cardiovascular suspects who were transferred by ambulance to the emergency department (ED) between 1 October 2019 and 30 November 2021. Global community medical care was provided by the Public Health System (SACYL) and included the Emergency Coordination Center (1-1-2 phone backup), 6 advanced life support (ALS), 43 basic life support (BLS) and 5 hospitals (one minor general district hospital and four university tertiary hospitals). Normally, BLS is staffed by two emergency medical technicians (EMTs), performing on-scene or *en route* BCLS work-up protocols, and ALS is made up of two EMTs, an emergency registered nurse (ERN) and a physician, conducting ACLS operations.

All cases were examined by an ALS, and following the assessment and diagnostic tests, the physician determined, in line with current guidelines and according to the individual clinical situation, the need for transfer to the ED as well as the type of ambulance: BLS or ALS. All hospitals presented the acute cardiac care unit (ACCU) and three hospitals presented the cardiac intervention room 24×7 and emergency cardiac surgery unit. Patients who needed emergency haemodynamic studies or advanced cardiologic care and for whom these facilities were not available at the reference hospital were evacuated as top priority (daytime by Helicopter Emergency Medical Service and nighttime by ALS, mandatory with turnaround times under 1 hour) to other hospitals included in the study.

### Population
Recruitment was consecutive. Participants enrolled in the study were defined as adults (>18 years) with acute cardiovascular disease (prehospital syndromic suspected condition) managed by EMS and transferred to the ED. Non-cardiovascular disorders, minors, pregnant women (known or apparent), terminally ill patients (condition confirmed by a medical report) or on-site discharge were excluded.

An informed consent form, managed by the ERN and applicable for the entire follow-up study, was reviewed and countersigned by all participants. In the absence of appropriate understanding by the patient, a research associate in the ED tried to collect the consent form signed by the patient or by a family member or legal guardian. If, despite all efforts, consent was not obtained, the patient was removed and excluded from the study.

## Data collection

Mandatory on-site hands-on training was conducted for all staff before the study started and included the standardised procedure for taking vital signs, handling, calibration and cleaning of the point-of-care testing device as well as data input to a database specially created for this purpose. A specific database was designed, with access by individual passwords and double authentication. In this database, we entered both the data collected from the EMS medical records and later the data from hospital care and subsequent follow-up by reviewing the electronic medical records (in two steps, at 30 and 365 days). Once the data had been linked, the data manager anonymised the patient identifiers.

Age, sex, nursing home location and on-scene vital signs (respiratory rate, oxygen saturation, systolic and diastolic blood pressure, heart rate, temperature, Glasgow Coma Scale, glucose, lactate and ECG) were collected and recorded by the ERN. A LifePAK 15 monitor-defibrillator (Physio-Control, Inc., Redmond, USA) was applied to obtain oxygen saturation, blood pressure, heart rate and ECG. A ThermoScan PRO 6000 thermometer (Welch Allyn, Skaneateles Falls) was used to collect temperature, and, finally, the analyser epoc (Siemens Healthcare GmbH, Erlangen Germany) was employed to perform prehospital analysis.

The physician subsequently checked the ECG and recorded the baseline heart rhythm as well as the 17 comorbidity categories needed to calculate the Age-Charlson comorbidity index (ACCI), composed of myocardial infarction, congestive heart failure, peripheral vascular disease, stroke or transient ischaemic attack, dementia, chronic obstructive pulmonary disease, connective tissue disease peptic ulcer disease, mild liver disease, uncomplicated diabetes mellitus, hemiplegia, moderate to severe chronic kidney disease, diabetes mellitus with end-organ damage, localised solid tumour, leukaemia, lymphoma, moderate to severe liver disease, metastatic solid tumour and AIDS.

## Outcomes

The primary outcome was cumulative mortality, from prehospital care to 1 year of follow-up, segregating time periods as follows: 1, 2, 7, 30, 90, 180 and 365 days. In addition, a comparison was performed matching two age-typed cohorts, a group of ≤74 years versus a group of ≥75 years, with age discrimination in line with similar reports.[17 18]

The secondary prehospital outcomes included advanced airway management (non-invasive or invasive mechanical ventilation), electrical therapy (transcutaneous pacemaker, cardioversion or defibrillation) and/or vasoactive agents. Finally, the ALS physician appointed the prehospital syndromic suspected condition, involving ischaemic heart disease, acute heart failure, arrhythmia, syncope or hypertensive emergency.

The secondary hospital outcomes were collected from the electronic medical records obtained at the 1-year follow-up and comprised cumulative mortality (all-cause), admission rate, echocardisocopy, percutaneous interventional vascular surgery, emergent surgery, advanced airway management, vasoactive agents and ACCU admission.

## Data analyses

Descriptive results and the associations between age and cardiovascular diagnosis with the variables were assessed by the Mann-Whitney U test or the $\chi^2$ test, when appropriate, and the effect size in the form of standardised mean difference was provided. Absolute values and percentages were used for categorical variables, and median IQRs were used for continuous variables because they did not follow a normal distribution. The procedure to determine those variables associated with mortality was as follows: first, a log-rank univariate analysis was performed. Then, a Cox regression (which included only those variables with $p<0.001$ in the log-rank univariate analysis) was performed to evaluate the association of demographic and prehospital parameters with mortality. The Cox regression results were expressed as the HR and 95% CI. Furthermore, survival according to age, cardiovascular diagnosis and the combination of both variables was obtained using the Kaplan-Meier method. The cumulative mortality by prehospital syndromic suspect and the difference between groups were assessed by the $\chi^2$ test. Finally, the descriptive statistics and association

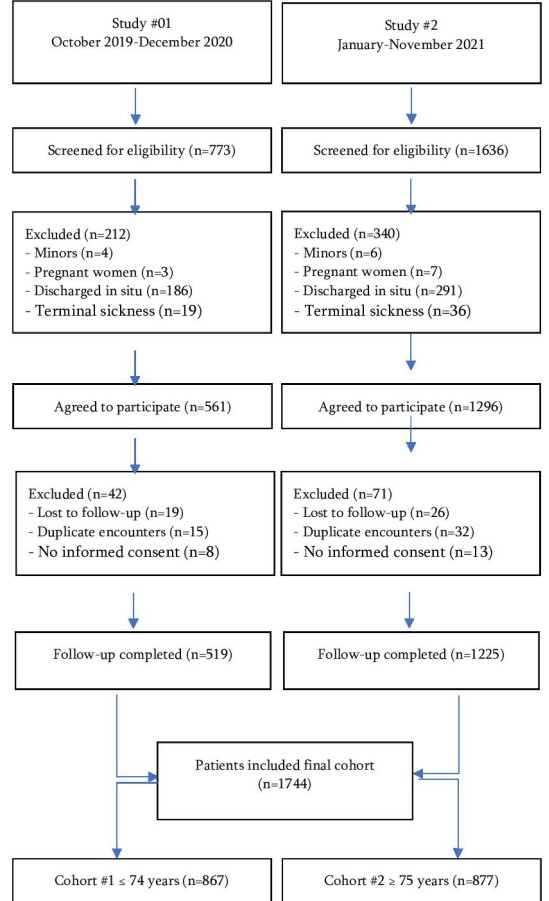

**Figure 1** Study population flowchart.

**Table 1** Demographic and clinical baseline variables

| Variable | ≤ 74 years | ≥ 75 years | Standardised difference† | Odds ratio‡ (95% CI) | P value§ |
|---|---|---|---|---|---|
| Number (%) with data* | 867 (49.7) | 877 (50.3) | NA | NA | NA |
| Epidemiological variables | | | | | |
| Sex, female | 287 (33.1) | 438 (49.9) | 0.168 | 0.50 (0.41 to 0.60) | <0.001 |
| Age, year | 62 (52-69) | 84 (79-88) | 2.486 | NA | <0.001 |
| Nursing homes | 21 (2.4) | 169 (19.3) | 0.169 | 0.10 (0.06 to 0.16) | <0.001 |
| On-scene vital signs | | | | | |
| RR, breaths/min | 16 (14-19) | 17 (14-23) | 0.166 | 0.98 (0.97 to 0.99) | 0.001 |
| $SpO_2$, % | 97 (95-98) | 96 (93-98) | −0.225 | 1.03 (1.02 to 1.04) | <0.001 |
| SBP, mm Hg | 135 (112-153) | 134 (112-157) | 0.043 | 1.00 (1.00 to 1.00) | 0.366 |
| DBP, mm Hg | 80 (67-94) | 73 (60-86) | −0.301 | 1.02 (1.01 to 1.02) | <0.001 |
| HR, beats/min | 80 (65-100) | 78 (62-99) | −0.119 | 1.00 (1.00 to 1.01) | 0.013 |
| Temperature, °C | 36 (35.9-36.5) | 36 (35.8-36.5) | 0.013 | 0.98 (0.87 to 1.11) | 0.778 |
| Glasgow coma scale, points | 15 (15-15) | 15 (15-15) | −0.019 | 1.01 (0.97 to 1.05) | 0.691 |
| Glucose, mg/dL | 124 (105-151) | 141 (116-183) | 0.266 | 1.00 (0.99 to 1.00) | <0.001 |
| Lactate, mmol/L | 1.77 (1.1-2.82) | 1.97 (1.19-3.1) | 0.042 | 0.98 (0.94 to 1.02) | 0.376 |
| Baseline cardiac rhythm | | | | | |
| Sinus | 487 (56.2) | 338 (38.5) | −0.176 | 2.04 (1.69 to 2.47) | <0.001 |
| Atrial fibrillation | 127 (14.6) | 293 (33.4) | 0.187 | 0.34 (0.27 to 0.43) | <0.001 |
| Atrial flutter | 7 (0.8) | 8 (0.9) | 0.001 | 0.89 (0.30 to 2.52) | 0.820 |
| Atrial tachycardia | 98 (11.3) | 63 (7.2) | −0.041 | 1.64 (1.18 to 2.30) | 0.003 |
| Supraventricular tachycardia | 23 (2.7) | 12 (1.4) | −0.012 | 1.95 (0.98 to 4.10) | 0.058 |
| Ventricular tachycardia | 22 (2.5) | 4 (0.5) | −0.020 | 5.50 (2.08 to 19.3) | <0.001 |
| Sinus bradycardia | 62 (7.2) | 57 (6.5) | −0.006 | 1.11 (0.76 to 1.61) | 0.591 |
| 1° degree block | 3 (0.3) | 24 (2.7) | 0.023 | 0.13 (0.03 to 0.37) | <0.001 |
| 2° block type I | 2 (0.2) | 2 (0.2) | −0.000 | 1.01 (0.11 to 9.73) | 0.991 |
| 2° block type II | 4 (0.5) | 7 (0.8) | 0.003 | 0.59 (0.15 to 2.00) | 0.397 |
| Complete block | 16 (1.8) | 24 (2.7) | 0.008 | 0.67 (0.35 to 1.27) | 0.219 |
| Pacemaker | 6 (0.7) | 40 (4.6) | 0.008 | 0.15 (0.06 to 0.33) | <0.001 |
| Junctional | 1 (0.1) | 3 (0.3) | 0.038 | 0.37 (0.01 to 3.16) | 0.380 |
| Idioventricular | 3 (0.3) | 0 | −0.003 | NA | |
| Asystole | 3 (0.3) | 0 | −0.003 | NA | |
| Ventricular fibrillation | 3 (0.3) | 2 (0.2) | −0.001 | 1.48 (0.23 to 12.8) | 0.678 |
| Comorbidities | | | | | |
| ACCI, points | 1 (0–3) | 3 [2–5] | 0.557 | 0.79 (0.76 to 0.83) | <0.001 |
| Congestive heart failure | 109 (12.6) | 279 (31.8) | 0.192 | 0.31 (0.24 to 0.39) | <0.001 |
| Myocardial infarction | 237 (27.3) | 321 (36.6) | 0.092 | 0.65 (0.53 to 0.80) | <0.001 |
| Peripheral vascular disease | 97 (11.2) | 137 (15.6) | 0.044 | 0.68 (0.51 to 0.90) | 0.007 |
| Cerebrovascular disease | 37 (4.3) | 120 (13.7) | 0.094 | 0.28 (0.19 to 0.41) | <0.001 |
| Hemiplegia | 23 (2.7) | 48 (5.5) | 0.028 | 0.47 (0.28 to 0.78) | 0.003 |
| Chronic pulmonary disease | 189 (21.8) | 199 (22.7) | 0.008 | 0.95 (0.76 to 1.19) | 0.655 |
| DM uncomplicated | 113 (13) | 151 (17.2) | 0.041 | 0.72 (0.55 to 0.94) | 0.015 |
| DM end organ damage | 73 (8.4) | 129 (14.7) | 0.062 | 0.53 (0.39 to 0.72) | <0.001 |
| Moderate-severe CKD | 41 (4.7) | 190 (21.7) | 0.169 | 0.18 (0.13 to 0.25) | <0.001 |
| Mild hepatic disease | 30 (3.5) | 27 (3.1) | −0.003 | 1.13 (0.66 to 1.93) | 0.654 |
| Severe hepatic disease | 21 (2.4) | 16 (1.8) | −0.006 | 1.33 (0.69 to 2.62) | 0.387 |
| Peptic ulcer disease | 75 (8.7) | 76 (8.7) | 0.002 | 1.00 (0.71 to 1.40) | 0.991 |

Continued

del Pozo Vegas C, *et al. BMJ Open* 2023;**13**:e078815. doi:10.1136/bmjopen-2023-078815

**Table 1** Continued

| Variable | ≤ 74 years | ≥ 75 years | Standardised difference† | Odds ratio‡ (95% CI) | P value§ |
|---|---|---|---|---|---|
| AIDS | 4 (0.5) | 1 (0.1) | −0.003 | 3.68 (0.51 to 101) | 0.175 |
| Lymphoma | 7 (0.8) | 16 (1.8) | 0.010 | 0.44 (0.17 to 1.05) | 0.063 |
| Leukaemia | 13 (1.5) | 13 (1.5) | −0.002 | 1.01 (0.46 to 2.23) | 0.977 |
| Metastatic solid tumour | 20 (2.3) | 33 (3.8) | 0.014 | 0.61 (0.34 to 1.06) | 0.077 |
| Nonmetastatic solid tumour | 119 (13.7) | 209 (23.8) | 0.101 | 0.51 (0.40 to 0.65) | <0.001 |
| Connective tissue disease | 48 (5.5) | 49 (5.6) | 0.001 | 0.99 (0.661 to .49) | 0.963 |
| Dementia | 27 (3.1) | 136 (15.5) | 0.123 | 0.18(0.11 to 0.27) | <0.001 |

*Values expressed as total number (percentage) and medians (25th–75th percentile), as appropriate.
†The Mann-Whitney U test or chi-squared test was used as appropriate.
‡Cohen's d test was used to estimate the effect size.
§Fisher's exact probability statistic was used.
ACCI, Age-Charlson comorbidity index; AIDS, acquired immunodeficiency syndrome; CKP, chronic kidney disease; DBP, diastolic blood pressure; DM, diabetes mellitus; HR, heart rate; NA, not applicable; Ref, reference; RR, respiratory rate; SBP, systolic blood pressure; $SPO_2$, oxygen saturation.

of mortality at 1, 2, 7, 30, 90, 180 and 365 days for each prehospital syndromic suspect were assessed by univariate comparison and expressed as ORs and 95% CIs.

Data for prehospital covariates were prospectively collected and registered in a database generated with IBM SPSS Statistics for Apple version V.20.0 software (IBM, Armonk USA). The caseload entry system was tested to delete unclear or ambiguous items and to verify the adequacy of the data-gathering system. The data present missing values completely at random; therefore, the strategy used (listwise deletion) does not imply biased means, variances or regression weights. The statistical power (from 1 to 100) of the present study is 89.4 based on the following considerations: (1) the sample used is n=517, (2) significant level of p=0.05, (3) expected ORs of 0.405 and (4) 14% of events.

### Patient and public involvement

Patients and/or the public were not involved in the design, or conduct, or reporting or dissemination plans of this research.

### RESULTS

Among 2409 cases, 1744 patients with a syndromic cardiovascular suspect managed by EMS and transferred to the ED were finally included in the final analysis. There were 867 patients from cohort number 1 (≤74 years) and 877 from cohort number 2 (≥75 years), 49.7% versus 50.3% (figure 1).

Cohort number 1 was characterised by a median age of 62 years (IQR: 52–69) and 33.1% (287 cases) were women. Half of the cases (56.2%) showed sinus rhythm, and the comorbidity burden resulted in an ACCI of 1 point (IQR: 0–3), especially myocardial infarction (27.3%), chronic pulmonary disease (21.8%) and non-metastatic solid tumours (13.7%). In contrast, cohort number 2 was described by a median age of 84 years (IQR: 79–88), 49.9% (438 cases) were women, and one-fifth lived in

nursing homes. Despite sinus rhythm being the most common baseline cardiac rhythm (38.5%), the ratio of atrial fibrillation (33.4%) is noteworthy. The ACCI score was significantly higher, with a median of 3 points (IQR: 2–5) and a very pronounced prevalence of myocardial infarction (36.6%), congestive heart failure (31.8%) and non-metastatic solid tumours (23.8%) (table 1).

The 365-day cumulative mortality in the elderly amounted to 26.1% (229 cases) versus 11.6% (11.6%) in patients under 75 years old. Cohort number 1 presented a significantly increased percentage of mechanical ventilation, defibrillation, cardioversion, percutaneous interventional vascular surgery and ACCU admission versus cohort number 2, with a relatively increased incidence of non-invasive mechanical ventilation and transcutaneous pacemakers. Concerning prehospital syndromic suspected conditions, half of the patients under 75 years old presented with ischaemic heart disease (49%), followed by syncope (25.2%). The elderly also showed ischaemic heart disease (30.1%) and syncope (29.4%), stressing, in particular, the elevated incidence of acute heart failure (23.9% vs 8.8%, respectively, intercohort) (table 2). Cumulative mortality by prehospital syndromic suspected condition is reported in online supplemental table S1, as only those patients under 75 years presented statistically significant differences between the prehospital syndromic suspected conditions. The OR of each prehospital syndromic suspected condition for each mortality interval is found in online supplemental table S2 to S6.

The Cox regression (table 3) that included all those variables with p<0.001 in the long-rank analysis (online supplemental table S7) showed that being elderly, being in a nursing home, high respiratory rate, low systolic blood pressure, high levels of lactate, use of non-invasive mechanical ventilation, mechanical ventilation, cardioversion defibrillation, suffering from acute heart failure and an elevated ACCI were variables statistically

**Table 2** Primary and key secondary outcomes

| Variable | ≤74 years | ≥75 years | Standardised difference† | Odds ratio‡ (95% CI) | P value§ |
|---|---|---|---|---|---|
| Number (%) with data* | 867 (49.7) | 877 (50.3) | NA | NA | NA |
| Cumulative mortality | | | | | |
| 1 day | 25 (2.9) | 35 (4) | 0.011 | 0.72 (0.42 to 1.20) | 0.205 |
| 2 days | 30 (3.5) | 51 (5.8) | 0.023 | 0.58 (0.36 to 0.92) | 0.019 |
| 7 days | 42 (4.8) | 69 (7.9) | 0.030 | 0.60 (0.40 to 0.88) | 0.010 |
| 30 days | 53 (6.1) | 106 (12.1) | 0.059 | 0.47 (0.33 to 0.67) | <0.001 |
| 90 days | 74 (8.5) | 156 (17.8) | 0.092 | 0.43 (0.32 to 0.58) | <0.001 |
| 180 days | 88 (10.1) | 184 (21) | 0.108 | 0.43 (0.32 to 0.56) | <0.001 |
| 365 days | 101 (11.6) | 229 (26.1) | 0.144 | 0.37 (0.29 to 0.48) | <0.001 |
| Secondary outcome (support on-scene) | | | | | |
| NIMV | 22 (2.5) | 65 (7.4) | 0.048 | 0.33 (0.20 to 0.53) | <0.001 |
| Mechanical ventilation | 39 (4.5) | 23 (2.6) | −0.018 | 1.74 (1.04 to 2.99) | 0.034 |
| Transcutaneous pacemaker | 27 (3.1) | 33 (3.8) | 0.006 | 0.82 (0.49 to 1.38) | 0.458 |
| Cardioversion | 22 (2.5) | 11 (1.3) | −0.012 | 2.03 (1.00 to 4.41) | 0.049 |
| Defibrillation | 24 (2.8) | 8 (0.9) | −0.018 | 3.05 (1.41 to 7.35) | 0.004 |
| Vasoactive agents | 31 (3.6) | 29 (3.3) | −0.002 | | 0.758 |
| Prehospital syndromic suspected condition | | | | | |
| Ischaemic heart disease | 425 (49) | 264 (30.1) | −0.189 | 0.45 (0.37 to 0.55) | <0.001 |
| Acute heart failure | 76 (8.8) | 210 (23.9) | 0.152 | 3.32 (2.51 to 4.43) | <0.001 |
| Arrhythmia | 112 (12.9) | 116 (13.2) | 0.001 | 1.02 (0.77 to 1.34) | 0.961 |
| Syncope | 221 (25.2) | 258 (29.4) | 0.039 | 1.22 (0.99 to 1.50) | 0.074 |
| Hypertensive emergency | 33 (3.8) | 29 (3.3) | −0.005 | 0.86 (0.52 to 1.44) | 0.664 |
| Hospital outcome | | | | | |
| Inpatient | 424 (48.9) | 487 (55.5) | 0.066 | 1.30 (1.08 to 1.58) | 0.006 |
| Hospitalisation time, days | 1 (0–6) | 2 (0–7) | 0.024 | 1.00 (0.99 to 1.01) | 0.613 |
| Echocardisocopy | 335 (38.6) | 303 (34.5) | −0.040 | 0.84 (0.69 to 1.02) | 0.076 |
| Fibrinolysis | 26 (3) | 10 (1.1) | −0.018 | 0.38 (0.17 to 0.77) | 0.006 |
| PIVS | 249 (28.7) | 157 (17.9) | −0.108 | 0.54 (0.43 to 0.68) | <0.001 |
| Emergent surgery | 21 (2.4) | 18 (2.1) | −0.003 | 0.85 (0.44 to 1.60) | 0.602 |
| NIMV | 26 (3) | 70 (8) | 0.049 | 2.79 (1.78 to 4.51) | <0.001 |
| Mechanical ventilation | 65 (7.5) | 36 (4.1) | −0.033 | 0.53 (0.34 to 0.80) | <0.001 |
| Vasoactive agents | 63 (7.3) | 55 (6.3) | −0.010 | 0.85 (0.59 to 1.24) | 0.002 |
| ACCU admission | 268 (30.9) | 149 (19) | −0.139 | 0.46 (0.36 to 0.57) | <0.001 |

*Values expressed as total number (percentage) and medians (25th-75th percentile), as appropriate.
†The Mann-Whitney U test or Mann-Whitney U test or chi-squared test was used as appropriate.
‡Cohen's d test was used to estimate the effect size.
§Fisher's exact probability statistic was used.
ACCU, acute cardiac care unit; NA, not applicable; NIMV, non-invasive mechanical ventilation; PIVS, percutaneous interventional vascular surgery; Ref, reference.

significantly associated with mortality. The aforementioned results for elderly individuals and those with acute heart failure are illustrated in figure 2. Figure 2A shows the survival curve of patients ≤74 years versus ≥75 years, with statistically significant differences (p<0.001) that appeared at the beginning of the follow-up and remained stable over time. Figure 2B shows the mortality curves for each prehospital syndromic suspected condition, with acute heart failure presenting the highest mortality that started at the beginning of the follow-up and remained

stable over time. When analysing the results according to age, those patients suffering from arrhythmia and syncope presented statistically significant differences in the mortality curves (online supplemental figure S1) but not the other conditions. This was also the case when considering the association of each particular condition with mortality at 1, 2, 7, 30, 90, 180 and 365 days (online supplemental table S2 to S6); only arrhythmia and syncope presented statistically significant time points,

**Table 3**  HR derived from Cox regression

| Variable | HR | 5% CI | 95% CI | P value |
|---|---|---|---|---|
| Cohort≥75 years | 1.98 | 1.51 | 2.60 | <0.001 |
| Being in nursing home | 1.42 | 1.07 | 1.89 | 0.014 |
| Respiratory rate | 1.02 | 1.01 | 1.04 | <0.001 |
| Systolic blood pressure | 0.99 | 0.98 | 0.99 | 0.017 |
| Lactate | 1.25 | 1.20 | 1.30 | <0.001 |
| Non-invasive mechanical ventilation | 1.83 | 1.26 | 2.65 | 0.001 |
| Mechanical ventilation | 2.53 | 1.30 | 4.92 | 0.006 |
| Cardioversion | 2.21 | 1.22 | 4.03 | 0.009 |
| Defibrillation | 2.39 | 1.23 | 4.64 | 0.010 |
| Acute heart failure | 2.32 | 1.63 | 3.31 | <0.001 |
| Age-Charlson comorbidity index | 1.13 | 1.09 | 1.18 | <0.001 |

arrhythmia only at 365-day mortality, and syncope at all time points except for 1-day mortality.

## DISCUSSION

This prospective, multicentre, observational, EMS delivery study assessed the association of prehospital variables, particularly elderly variables, with short-term and long-term mortality in patients with acute life-threatening cardiovascular diseases. Elderly patients (≥75 years) presented a twofold risk of mortality compared with patients ≤74 years. Life-threatening interventions (mechanical ventilation, cardioversion and defibrillation) were also related to a twofold increased risk of mortality. Importantly, patients suffering from acute heart failure presented a more than fold increased risk of mortality.

Cardiovascular diseases are the leading cause of EMS attendance[1] and are associated with high morbidity and mortality.[19] Therefore, the accurate and quick characterisation of patients at the first contact could provide critical information for the development of specialised risk scores[20] or specific detection codes.[4] In this sense, the present study aims to describe the association of prehospital variables with mortality, particularly in elderly individuals, which is one of the main risk factors for cardiovascular diseases.[21] Our results align with this evidence, since the categorisation of patients ≤74 years and ≥75 years revealed that older patients presented a higher rate of mortality. Moreover, other factors related to elderly individuals, such as being in nursing homes or the number of comorbidities, measured by the ACCI, were risk factors for mortality. As expected, the higher the number of comorbidities, the higher the risk of mortality. This is not surprising since several comorbidities worsen cardiovascular conditions.[14] In fact, comorbidities make patient examination difficult, which is particularly true in the prehospital setting.[22] Strikingly, the leap to worse long-term mortality outcomes occurs above 75 years old, whereas below this age, short-term mortality outcomes take prominence.

Despite all these evidence, the effect of age on mortality was not found for short-term mortality, and no statistically significant effect was found for 1-day or 2-day mortality; instead, the statistical significance was relevant at 30 days onwards. This short-term mortality result is somewhat surprising, since it has been previously described at the prehospital level that older ages are associated with higher short-term mortality.[23] Perhaps this difference could be explained by the fact that those patients were characterised as suffering mainly from respiratory diseases.[23] The

A
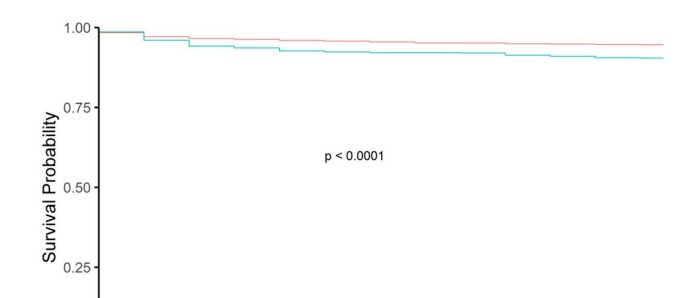

B
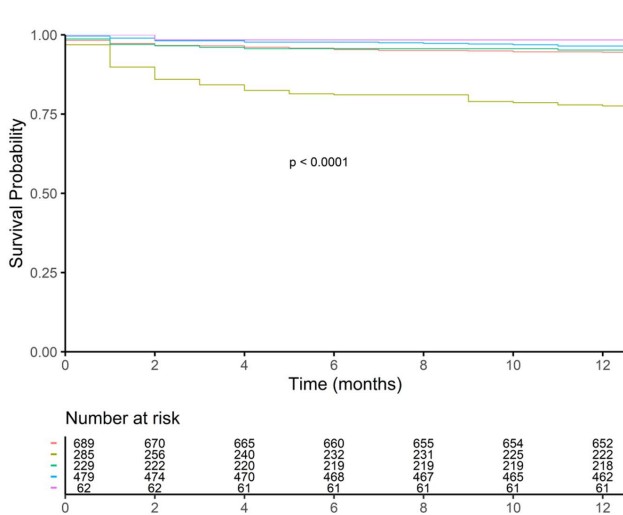

**Figure 2**  Kaplan-Meier survival curves for age≤74 years vs≥75 years (A) and cardiovascular diagnosis for both groups together (B). Age≤74 years (red line) and age≥75 years (blue line) (A). Ischaemic heart disease (red line), acute heart failure (dark green), arrhythmia (light green), syncope (blue line), hypertensive emergency (purple line) (B).

long-term mortality relationship with age has already been described in the prehospital setting.[24]

Lactate was statistically related to long-term mortality. The lactate value as a predictor of mortality is well documented in prehospital critical care and represents a very powerful indicator of mitochondrial hypoperfusion, directly affecting the production of bioavailable energy for all physiological processes, including the cardiovascular system.[6] This has also been reported, as for respiratory rate and age, for long-term mortality when dealing with all patients and not with disease-specific analysis.[24] Enriquez de Salamanca Gambara *et al*[24] also found that GCS and SpO2 were associated with mortality. Perhaps GCS was more related to neurological diseases than cardiovascular diseases, which is our case. A similar argument could be made for SpO2, which could be related to respiratory diseases.

Life-threatening interventions are closely related to patients suffering from cardiovascular disease,[25] which was also our case, since non-invasive mechanical ventilation, mechanical ventilation, cardioversion and defibrillation were procedures present in the non-survivor group. Patients who must undergo ALS interventions on scene or en route, even overcoming the situation that originated the life-threatening intervention, are also negatively impacted in the long term. In other words, in patients needed to receive more aggressive manoeuvres, clinical evolution should be considered, even more so in elderly and frail patients.

Particularly relevant is the fact that the mortality of each prehospital syndromic suspected condition was different when considering elderly individuals. As expected, acute heart failure was the main condition related to death; however, when analysing mortality according to age groups, only arrhythmia and syncope presented statistically significant differences. This should be interpreted as the higher the age, the higher the probability of death, but only for those two conditions. The other conditions should be treated independently of age. The results for syncope and arrhythmia could be explained by the increased incidence with age, particularly at 70 years for syncope[26] and arrhythmias, especially for atrial fibrillation.[27]

Our study has some strengths, including the sample size, novelty, and study design with a reduced loss-to-follow-up rate. It was conducted in both rural and urban areas, and our results could be generalisable to other health systems. This last point is based on the fact that all parameters associated with mortality can be easily accessed by the EMS staff. Only lactate determination will require a point-of-care device, which is now a reality in several EMSs.[28] This generalisability could lead to scores or specific detection codes. However, some limitations must be considered. First, this is an observational study, so we cannot rule out the possibility of selection bias, although participating centres had previous experience and enrolled consecutive patients with prehospital acute cardiovascular disease. To obtain a representative sampling, cases were collected 24/7/365 non-stop in urban, semiurban and rural areas and in different ambulance stations. Second, possible bias may exist in relation to case inclusion in the study. All prehospital acute cardiovascular conditions were included; however, certain disorders may raise uncertainty, that is, stomachache could be labelled a digestive disease at prehospital care, although following complementary in-hospital tests (imaging studies, laboratory tests, etc), the disease ends up being categorised as ischaemic heart disease. Nevertheless, the limitation was dampened for two reasons. On-scene, ALS physicians could issue up to a maximum of three diagnostic suspicions, and, therefore, any case with a diagnosis of acute cardiovascular disease was eventually included in the analysis. In addition, the diagnostic coincidence between prehospital syndromic cardiovascular suspect (ischaemic heart disease, acute heart failure, arrhythmia, syncope and hypertensive emergency) and the final in-hospital diagnosis was very consistently strong. Third, data extractors were not blinded. To avoid cross-contamination, EMS providers lacked access to hospital follow-up data, and hospital investigators remained blinded to prehospital care data. To accurately link the data between prehospital care and hospital follow-up, at least five of the following extractors had to be an exact match: health system ID card, incident reference (EMS-register), first and last name, age, sex at birth, date and/or time of arrival at the ED. Total access to the master database was given exclusively by the principal investigator and data manager. Fourth, modifiable lifestyles (smoking, alcohol consumption, weight, diet quality and physical activity) or medication history are indeed associated with short-term and long-term mortality in life-threatening acute cardiovascular disease. Nevertheless, these data could not be collected in prehospital care or during the in-hospital follow-up phase. For subsequent investigations, a postindex event interview procedure will be implemented to collect these data and, thus, be able to analyse the influence of these covariates on the final outcome. Finally, an adequate sample size was used in the present preliminary study, but multicentre studies in different health systems are needed to confirm the generalisability of the results.

In summary, this study revealed the prehospital variables associated with the long-term mortality of patients suffering from acute cardiovascular disease. Elderly age, life-threatening interventions, acute heart failure, comorbidities and lactate should be considered when EMS patients have acute life-threatening cardiovascular diseases. In particular, the effect on elderly patients presenting arrhythmia or syncope should be considered. Finally, our results could pave the way for the development of specific codes or scores for cardiovascular diseases to facilitate the risk of mortality characterisation.

**Author affiliations**
¹Emergency Department, Hospital Clinico Universitario de Valladolid, Valladolid, Castilla y León, Spain
²Universidad de Valladolid, Valladolid, Spain
³University of Castilla-La Mancha—Center for University Studies Talavera de la Reina, Talavera de la Reina, Castilla-La Mancha, Spain
⁴Hosp Univ Rio Hortega, Valladolid, Spain
⁵CIBER of Respiratory Diseases, Instituto de Salud Carlos III, Madrid, Spain
⁶Advanced Life Support, Emergency Medical Services (SACYL), Valladolid, Spain
⁷Universidad Europea del Atlántico, Santander, Spain
⁸Universidad Internacional Iberoamericana, Arecibo, Puerto Rico, USA
⁹Universidad de La Romana, La Romana, Dominican Republic
¹⁰Universidad Internacional Iberoamericana, Campeche, Mexico
¹¹Universidade Internacional do Cuanza, Cuito, Bié, Angola
¹²Fundación Universitaria Internacional de Colombia, Bogotá, Colombia

**Contributors** FM-R conceptualised the project, managed and coordinated the project, assisted with the design of methodology, analysed data and prepared the initial and final drafts of the manuscript. AS-G took responsibility for the data and their analysis. CdPV, DZ-S, SS-B and RL-I assisted with the management and coordination of the project, assisted with the design of the methodology, and helped review the manuscript. AS-G is responsible for the overall content as the guarantor. All authors performed a critical review and approved the final manuscript for interpretation of the data and important intellectual input.

**Funding** This work was supported by the Gerencia Regional de Salud, Public Health System of Castilla y León (Spain) [grant numbers GRS 1903/A/19 and GRS 2131/A/20].

**Competing interests** All signing authors meet the requirements of authorship and have declared the nonexistence of potential conflicts of interest. DZ-S, AS-G, CdPV, RL-I, SS-B and FM-R report no conflicts of interest. The authors have no disclosures to make. On behalf of the other authors, the corresponding author guarantees the accuracy, transparency and honesty of the data and information contained in the study, that no relevant information has been omitted and that all discrepancies between authors have been adequately resolved and described.

**Patient and public involvement** Patients and/or the public were not involved in the design, or conduct, or reporting, or dissemination plans of this research.

**Patient consent for publication** Not applicable.

**Ethics approval** Ethical approval was granted by the Rio Hortega University Hospital Ethics Committee, references PI041-19 and PI217-20 (principal investigator's reference centre). This authorisation applies to the other locations covered by the study. Participants gave informed consent to participate in the study before taking part.

**Provenance and peer review** Not commissioned; externally peer reviewed.

**Data availability statement** Data are available upon reasonable request. The data that support the findings of this study are available on request from the corresponding author AS-G. The data are not publicly available due to restrictions, and their information could compromise the privacy of research participants.

**ORCID iD**
Ancor Sanz-Garcia http://orcid.org/0000-0002-5024-5108

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
