## [Reviewer comments · BMJ Open]

ARTICLE DETAILS

TITLE (PROVISIONAL)	Prehospital acute life-threatening cardiovascular disease in elderly: An observational, prospective, multicenter, ambulance-based cohort study.
AUTHORS	del Pozo Vegas, Carlos; Zalama-Sánchez, Daniel; Sanz-Garcia, Ancor; López-Izquierdo, Raúl; Sáez-Belloso, Silvia; Oleaga, Cristina Mazas Perez; Azpíroz, Irma Domínguez; Pascual, Iñaki Elío; Martín-Rodríguez, Francisco

VERSION 1 – REVIEW

REVIEWER	Moghaddam, Nader Markazi Shahid Beheshti University of Medical Sciences
REVIEW RETURNED	01-Sep-2023

GENERAL COMMENTS	This article aims to explore the association of demographic and prehospital parameters with short- and long-term mortality in acute life-threatening cardiovascular disease. I personally think it is a very important issue and I just give some personal suggestions to help the authors improve their manuscript and increase the quality of their presentation. 1. In the title and aim of the study, the authors focused on the elderly by comparing two age cohorts: under 75 years versus over 75 years. In addition, it is mentioned that a comparison was performed matching two age-typed cohorts, a group of ≤ 74 years vs a group of ≥ 75 years, with age discrimination in line with similar reports. Although, there is no standard criterion to tell that an old person is considered elderly. But, in Western countries, an elderly person is defined as someone who has reached the chronological age of 65 years old. The stage is further divided into two sections; those that lie between the ages of 65 to 74 years old called the early elderly, and those that are over the age of 75 years who are referred to as late elderly. However, the same criteria don't apply in every medical setting, and some people in their mid-80s are also considered elderly. In some similar studies, they have considered elderly who has reached age ≥ 70 years old. Also, WHO had done new research recently, according to average health quality and life expectancy, and defined a new criterion that divides human age as follows: • 0-17 years old: underage• 18-65 years old: youth/young people• 66-79 years old: middle-aged• 80-99 years old: elderly/senior• 100+ years old: long-lived elderly So, it seems that over the age of 65 or 80 years old can be considered elderly. It is suggested to explain the reason of that criterion for elderly in the introduction and emphasize in the aim of the study.
--

	2. Some important risk factors such as modifiable lifestyle (cigarette smoking, alcohol consumption, weight, diet quality and physical activity), past medication history and use of some medications (i.e., anti hypertensives, antiplatelet agents and ...) are positively associated with short- and long-term mortality in acute life-threatening cardiovascular disease and have a confounding role which were not included in this study, and it's not determined whether these factors have been collected or not. 3. The above-mentioned factors should be considered as covariates in adjusted model if those exist in the database. If those are not existed, it should be explained as a limitation in discussion section. 4. Overall, the authors should effort to enhance scientific rigor through more explanation of research aim, methodology, analysis and interpretation.
--	--

REVIEWER	Higashikawa, Toshihiro Kanazawa Medical University, geriatric medicine
REVIEW RETURNED	03-Sep-2023

GENERAL COMMENTS	Dear Authors Throughout review,I found this study very interesting. This study is exploring the mortality in patients with cardiovascular diseases. Potential compounding factors only in cardiovascular disease are not excluded. There is a lack of important information. In particular, the elderly patients have a comorbidity,i.e., nutritional status, and physical deterioration,which influence the mortality. You should address the numbers and types of cardioprotective agents,i.e, SGLT2-inhibitors, ARBs, diuretics.Including these parameters ,you should perform cox regression analysis and extend discussion extensively.
---

VERSION 1 – AUTHOR RESPONSE

REVIEWER # 1

1. In the title and aim of the study, the authors focused on the elderly by comparing two age cohorts: under 75 years versus over 75 years. In addition, it is mentioned that a comparison was performed matching two age-typed cohorts, a group of ≤ 74 years vs a group of ≥ 75 years, with age discrimination in line with similar reports.

Although, there is no standard criterion to tell that an old person is considered elderly. But, in Western countries, an elderly person is defined as someone who has reached the chronological age of 65 years old. The stage is further divided into two sections; those that lie between the ages of 65 to 74 years old called the early elderly, and those that are over the age of 75 years who are referred to as late elderly. However, the same criteria don't apply in every medical setting, and some people in their mid-80s are also considered elderly.

In some similar studies, they have considered elderly who has reached age \geq 70 years old. Also, WHO had done new research recently, according to average health quality and life expectancy, and defined a new criterion that divides human age as follows: • 0-17 years old: underage • 18-65 years old: youth/young people • 66-79 years old: middle-aged • 80-99 years old: elderly/senior • 100+ years old: long-lived elderly

So, it seems that over the age of 65 or 80 years old can be considered elderly. It is suggested to explain the reason of that criterion for elderly in the introduction and emphasize in the aim of the study.

ANSWER:

We could not agree with the reviewer. Marking the 75-year line was a challenge in deciding the best time cut-off. It was decided to use 75 years as a cluster differentiation, based on the consistency of the literature and the fact that cardiovascular diseases and their worst outcomes begin to appear from the age of 75 years onwards.

We have introduced a new paragraph (1) in the introduction explaining these terms, and in addition, before the objective, we have explained (2) why it was decided to consider 75 years as the time mark to analyse the different cohorts.

1-“ Defining the concept of older adults is complex, as there are different categories and timelines, and there is no standard criterion to say that an older person is considered elderly (9). However, there is a general acceptance that persons over 65 years of age are considered elderly, and persons over 75 years of age are classified as late elderly. On the other hand, life expectancy and ageing control have improved considerably, with age-related comorbidities usually appearing later, which is the main reason why the 75-year age limit has been selected to differentiate the cohorts of the present work (10).”

2-“ Nonetheless, at 75 years old cardiovascular pathology in particular, with considerable progression with significant increases in comorbidities, emergence of additional diseases and exacerbation of already prevalent pathologies. The group of elderly individuals over 75 years old constitutes a cluster of special follow-ups that could be especially worthwhile for an in-depth characterization (10) (15).”

2. Some important risk factors such as modifiable lifestyle (cigarette smoking, alcohol consumption, weight, diet quality and physical activity), past medication history and use of some medications (i.e., anti hypertensives, antiplatelet agents and ...) are positively associated with short- and long-term mortality in acute life-threatening cardiovascular disease and have a confounding role which were not included in this study, and it's not determined whether these factors have been collected or not.

ANSWER: In the following question, we included the response to this point.

3. The above-mentioned factors should be considered as covariates in adjusted model if those exist in the database. If those are not existed, it should be explained as a limitation in discussion section.

ANSWER: We agree that modifiable lifestyles (smoking, alcohol consumption, weight, diet quality and physical activity) or medication history are indeed associated with short- and long-term mortality in life-threatening acute cardiovascular disease. However, during data collection in the prehospital setting, it was not possible to obtain these data due to patient condition and limited available time with the patient during attendance. Later, during the follow-up phase, it was not possible to retrieve this information from the electronic health record. Therefore, if the editor and the reviewer deem it appropriate, as a consensus solution, we propose introducing a new limitation stating these facts.

“Fourth, modifiable lifestyles (smoking, alcohol consumption, weight, diet quality and physical activity) or medication history are indeed associated with short- and long-term mortality in life-threatening acute cardiovascular disease. Nevertheless, these data could not be collected in prehospital care or during the in-hospital follow-up phase. For subsequent investigations, a postindex event interview procedure will be implemented to collect these data and thus be able to analyse the influence of these covariates on the final outcome.”

4. Overall, the authors should effort to enhance scientific rigor through more explanation of research aim, methodology, analysis and interpretation.

ANSWER: All sections have been extended to enhance scientific rigor.

REVIEWER #2

1. Potential compounding factors only in cardiovascular disease are not excluded.

There is a lack of important information. In particular, the elderly patients have a comorbidity, i.e., nutritional status, and physical deterioration, which influence the mortality.

You should address the numbers and types of cardioprotective agents, i.e., SGLT2-inhibitors, ARBs, diuretics. Including these parameters, you should perform cox regression analysis and extend discussion extensively.

ANSWER: We agree with the reviewer, and as we have replied to reviewer #1, who raises the same question in very similar terms, we have included a new limitation to describe this problem.

“Fourth, modifiable lifestyles (smoking, alcohol consumption, weight, diet quality and physical activity) or medication history are indeed associated with short- and long-term mortality in life-threatening acute cardiovascular disease. Nevertheless, these data could not be collected in prehospital care or during the in-hospital follow-up phase. For subsequent investigations, a postindex event interview procedure will be implemented to collect these data and thus be able to analyse the influence of these covariates on the final outcome.”

VERSION 2 – REVIEW

REVIEWER	Moghaddam, Nader Markazi Shahid Beheshti University of Medical Sciences
REVIEW RETURNED	20-Sep-2023
GENERAL COMMENTS	The manuscript has been revised according to the suggestions and comments.

VERSION 2 – AUTHOR RESPONSE